# 3D RNA-scaffolded wireframe origami

Molly F. Parsons [1,10], Matthew F. Allan[1,2,3,10], Shanshan Li [4,7,10],
Tyson R. Shepherd[1,8], Sakul Ratanalert [1,5,9], Kaiming Zhang[4,7],
Krista M. Pullen [1], Wah Chiu [4,6], Silvi Rouskin[2] & Mark Bathe [1] ✉

Hybrid RNA:DNA origami, in which a long RNA scaffold strand folds into a target nanostructure via thermal annealing with complementary DNA oligos, has only been explored to a limited extent despite its unique potential for biomedical delivery of mRNA, tertiary structure characterization of long RNAs, and fabrication of artificial ribozymes. Here, we investigate design principles of three-dimensional wireframe RNA-scaffolded origami rendered as polyhedra composed of dual-duplex edges. We computationally design, fabricate, and characterize tetrahedra folded from an EGFP-encoding messenger RNA and de Bruijn sequences, an octahedron folded with M13 transcript RNA, and an octahedron and pentagonal bipyramids folded with 23S ribosomal RNA, demonstrating the ability to make diverse polyhedral shapes with distinct structural and functional RNA scaffolds. We characterize secondary and tertiary structures using dimethyl sulfate mutational profiling and cryo-electron microscopy, revealing insight into both global and local, base-level structures of origami. Our top-down sequence design strategy enables the use of long RNAs as functional scaffolds for complex wireframe origami.

Nucleic acid nanotechnology offers unique capabilities for applications ranging from therapeutics[1–3] and enzyme nanoreactors[4–6] to computing[7,8] and materials synthesis[9–13]. The predictability of Watson-Crick-Franklin base-pairing together with crossover motifs derived from the Holliday junction renders nucleic acids amenable to fabricating a wide variety of 2D and 3D structures. Scaffolded DNA origami[14], in particular, has been demonstrated to enable the fabrication of nearly arbitrary 2D and 3D dense, bricklike[15–17] as well as porous, meshlike wireframe[18–24] objects by folding a single-stranded DNA scaffold to user-specified geometries via annealing with shorter DNA staple strands. These discrete DNA structures are versatile and broadly useful for numerous applications in materials[9,11,12,25], therapeutics[1,2] and cellular biophysics[3,26,27].

Incorporation of RNA in molecular origami with controlled target geometry offers additional applications in nanoscale materials synthesis and therapeutics that are not currently possible using purely DNA:DNA origami alone[28–31]. For example, modulation of immunostimulation by therapeutics or prophylactics[28–30,32]; tunable release kinetics via nuclease-specific degradation for nucleic acid or other payload delivery[1,28]; and organization of RNAs for structural and catalytic applications[5,33,34]. In addition, the use of RNA as a scaffold material also has the practical advantage over typical DNA scaffolds that naturally abundant long single-stranded RNA (ssRNA) such as ribosomal RNA (rRNA), highly prevalent in cells[35], may be used as a natural, albeit sequence-restricted source of scaffold, and scalable production of long single-stranded RNA with arbitrary sequence is also facile with

[1]Department of Biological Engineering, Massachusetts Institute of Technology, Cambridge, MA 02139, USA. [2]Department of Microbiology, Harvard Medical School, Boston, MA, USA. [3]Computational and Systems Biology, Massachusetts Institute of Technology, Cambridge, MA 02139, USA. [4]Department of Bioengineering, Stanford University, Stanford, CA 94305, USA. [5]Department of Chemical Engineering, Massachusetts Institute of Technology, Cambridge, MA 02139, USA. [6]CryoEM and Bioimaging Division, Stanford Synchrotron Radiation Lightsource, SLAC National Accelerator Laboratory, Stanford University, Menlo Park, CA 94025, USA. [7]Present address: MOE Key Laboratory for Cellular Dynamics and Division of Life Sciences and Medicine, University of Science and Technology of China, Hefei 230027, China. [8]Present address: Inscripta, Inc., Boulder, CO 80027, USA. [9]Present address: Department of Chemical and Biomolecular Engineering, Johns Hopkins University, Baltimore, MD 21218, USA. [10]These authors contributed equally: Molly F. Parsons, Matthew F. Allan, Shanshan Li. ✉e-mail: mark.bathe@mit.edu

in vitro transcription[36,37]. Finally, use of RNA scaffolds for origami enables base-level insight into secondary structure through chemical probing, offering insight into the impact of sequence design rules on local structural stability that is of significant interest to nucleic acid nanostructure design generally[38–40].

Unlike DNA, RNA carries an additional 2′-hydroxyl group on the sugar that forces a C3′-endo form, leading to an A-form double helix (11 base pairs per helical turn, 2.6 Å axial rise per base pair, 23 Å helical diameter), when hybridized with either DNA or RNA, rather than the canonical B-form of duplex DNA (10.5 base pairs per helical turn, 3.4 Å axial rise per base pair, 20 Å helical diameter)[41]. In addition, RNA is typically single-stranded in the cell, adopting complex tertiary folds with Hoogsteen and sugar edge base interactions, thereby rendering reliable de novo tertiary structure prediction and associated structural programmability challenging[42,43]. Nevertheless, knowledge gained from RNA tertiary structures has been used to generate RNA nanoparticles by engineering RNA fragments to self-assemble into programmed higher-order geometries using, for example, tRNAs and multi-way junctions to create complex shapes[44–48]. However, absolute control over the programmability of the final, target shape with this approach is constrained by the requisite sequences of the underlying folds.

As an alternative to the preceding tectonics approach[44,46,47] that leverages native RNA tertiary structures, tile-based assembly and RNA-scaffolded origami require specifically programming secondary structure to self-assemble target tertiary structures. Multicomponent tile-based assembly has been employed with either RNA:RNA or RNA:DNA interactions to form wireframe structures with single duplex[10,13,28,32,49–51] or dual-duplex edges[52]. Other recent advances in programming structures have focused on enabling co-transcriptional folding using a single RNA strand that folds isothermally into the target 2D structure that may include predefined tertiary junctions such as kissing loops[53–56]. Bottom-up design of RNA-scaffolded origami has been demonstrated using RNA staples to form 2D structures[57] and single-duplex-edged 3D wireframe structures[58], and using DNA staples to form bricklike[59] or 2D objects[57,60,61]. However, automated sequence design algorithms, which have greatly aided the dissemination of

scaffolded DNA origami[18,20–23], remain sparse[55,62], and no studies have yet realized the design and fabrication of arbitrary dual-duplex wireframe 3D structures with and RNA scaffolds of varying sequence and length.

Here, we explore design strategies for hybrid nucleic acid origami that use DNA oligo staples to fold in vitro transcribed RNA scaffolds of varying sequences and lengths into distinct 3D wireframe polyhedra with dual-duplex edges. Impacts of folding buffers, annealing protocols, and staple-to-scaffold ratios are first characterized with gel electrophoresis; target structures under suitable folding conditions are then validated using cryo-EM reconstruction. Secondary structure hybridization and stability are additionally assessed at single-nucleotide resolution using DMS-MaPseq, which provides insight into specific sources of structural instability in origami design. Our design strategy is distributed using a modified version of the open-source software DAEDALUS implementing A-form dual-duplex wireframe design rules.

## Results

### Sequence design and biochemical characterization of 3D RNA-scaffolded wireframe origami

We designed wireframe polyhedra with dual-duplex edges and anti-parallel crossovers (DX) that allow RNA to be used as the scaffold with DNA staples. To account for A-form duplex geometry in our hybrid RNA:DNA origami sequence design, we assumed 11 nt per helical turn and incorporated asymmetric spacing between adjacent crossovers of distinct strands[54,63], namely between adjacent scaffold and staple strand crossovers (Fig. 1).

Optimal folding conditions for origami typically vary based on design. For example, optimal annealing time and temperature as well as ionic species and concentrations differ between wireframe and bricklike designs, wireframe edge-types, and single-stranded RNA- vs. DNA-scaffolded origami[18,21,55,64]. For RNA:DNA dual-duplex wireframe origami, we adopted an analogous dual-duplex DNA wireframe origami annealing protocol[18], albeit reducing incubation time at high temperatures, omitting divalent cations (300 mM KCl instead of $MgCl_2$), and lowering the pH (10 mM HEPES pH 7.5 instead of 1x TAE pH

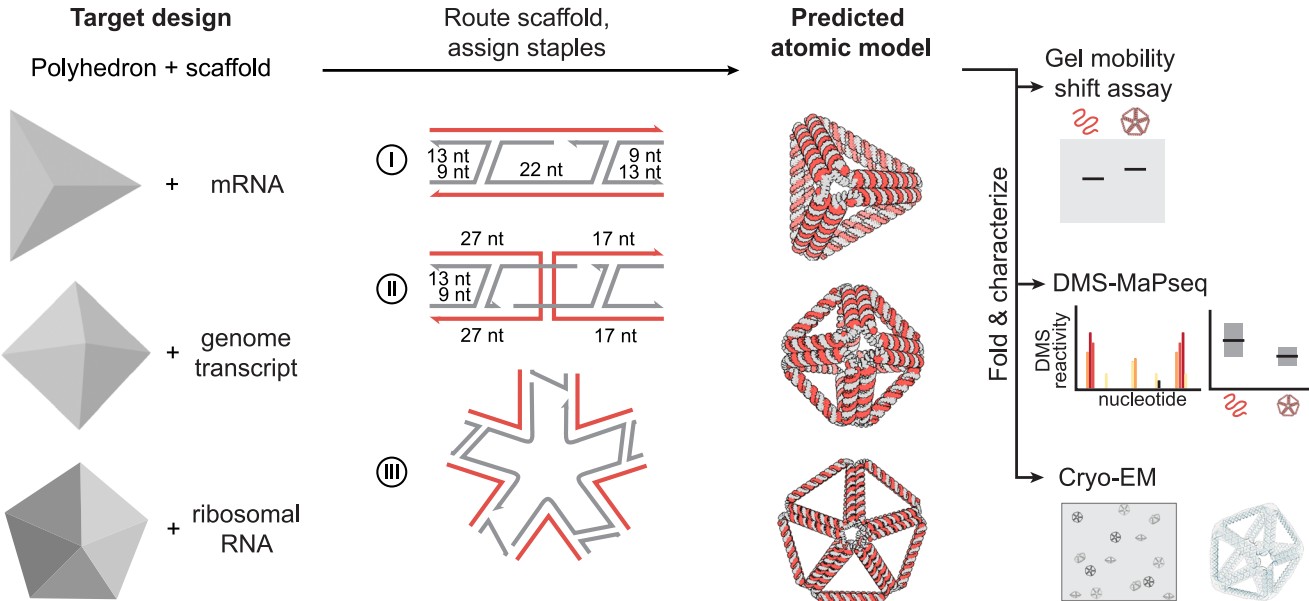

**Fig. 1 | Design overview for A-form DX wireframe origami.** Starting with a target polyhedron and scaffold sequence as inputs, we algorithmically route the scaffold through the polyhedron and route and assign staple for I. edges with no scaffold crossover, II. edges with a scaffold crossover, III. vertices, and predict an atomic model structure. The basic routing scheme for edges with 4 helical turns is shown,

scaffold in red and staples in grey. Using the calculated staple sequences, we can then fold the RNA scaffold into the target structure and characterize with gel mobility shift assays for preliminary folding evaluation, DMS-MaPseq or other RNA chemical probing method to evaluate base-pairing per nucleotide, and cryo-electron microscopy to evaluate overall structure formation.

8.0) to minimize RNA instability. We compared our selected protocol to two others employed in published RNA:DNA origami designs, one reported to fold 2D structures[61] and another to fold a 3D bricklike structure[59]. Gel mobility shift assays comparing these protocols as applied to two wireframe RNA:DNA polyhedra, an EGFP mRNA-scaffolded tetrahedron and a 23S rRNA-scaffolded pentagonal bipyramid with six helical turns (66 bp) per edge each, suggested that our selected protocol with 10 mM HEPES pH 7.5 + 300 mM KCl led to folded particles that were most compact. A fast folding protocol with a TAE + magnesium buffer reported for 2D RNA:DNA origami[61] led to a comparable gel mobility shift, but slightly more dimerization in the tetrahedron and a slightly more diffuse monomer band for the pentagonal bipyramid (Supplementary Fig. 1). The TE pH 7.5 + 40 mM NaCl buffer and overnight folding protocol reported for 3D bricklike RNA:DNA origami[59] led to a considerably greater mobility shift with a more diffuse band, indicating a less compactly folded structure for both wireframe particles characterized here (Supplementary Fig. 1). We thus concluded that the magnesium-free and higher monovalent salt buffer was an appropriate choice to fold these A-form wireframe RNA:DNA origami with DX edges. Titration of staple-to-scaffold ratios for folding the pentagonal bipyramid suggested that as little as a 2x molar ratio of DNA staples to RNA scaffold was sufficient for proper folding (Supplementary Fig. 2), consistent with a previous study of hybrid RNA:DNA origami that found for a 2D assembly that as little as 1x molar ratio of staples to scaffold was needed for proper folding[61], in contrast to the typical 10x excess used to fold DNA:DNA origami[18,19].

## Investigating polyhedral origami folding with distinct RNA scaffolds

To investigate the impact of RNA scaffold sequence on hybrid 3D wireframe origami folding, we examined the ability of three types of RNA sequences to scaffold A-form DX wireframe origami: an mRNA encoding bacterial EGFP; a De Bruijn sequence designed to have minimal self-complementarity and repetition (Methods); and an in vitro transcript of the M13 viral genome frequently used to scaffold DNA origami. For an initial test we targeted the design and fabrication of a regular tetrahedral geometry with six edges of equal length and four three-way vertices. We used in vitro-transcribed 792-nt prokaryotic EGFP mRNA and 660-nt and 924-nt De Bruijn RNA sequences to scaffold tetrahedra with six, five, and seven helical turns per edge, respectively, corresponding to 66, 55, and 77 bp per edge (rT66, rT55, and rT77).

A discrete gel mobility shift of folded origami relative to the scaffold suggested the rT66 formed a compactly folded particle (Fig. 2a). We applied dimethyl sulfate mutational profiling with sequencing (DMS-MaPseq)[65,66] to characterize folding biochemically with single base resolution, enabling the evaluation of the degree of base pairing between scaffold and staples. Mean normalized DMS reactivity for each of the $n = 64$ double helical segments in the folded origami, and corresponding segments of nucleotides in the EGFP mRNA scaffold alone, showed DMS reactivity was 82% lower ($P = 2.7 \times 10^{-10}$, two-sided Wilcoxon signed-rank test) in the folded origami (median = 0.63%) than in the scaffold without staples (median=3.4%) (Fig. 2b). Lower DMS reactivities in the origami were anticipated due to hybridized DNA staples protecting the RNA from DMS modifications. Refolding the rT66 in the absence of a specific single staple supported this interpretation (Supplementary Fig. 3), with the median DMS reactivity for the scaffold nucleotides targeted by this staple ~6.2-fold higher when the staple was omitted ($P = 1.4 \times 10^{-4}$, two-sided Wilcoxon signed-rank test). No such change was observed for off-target nucleotides (fold-change=1.08, $P = 0.76$), thereby also confirming hybridization specificity of this omitted staple. For the rT55 and rT77, gel mobility shift assays again indicated compactly folded particles (Supplementary Fig. 4a). Median DMS reactivity of the double-helical segments of each folded tetrahedron with a De Bruijn

scaffold sequence was also lower than that of its scaffold folded without staples (79% lower for rT55, 70% lower for rT77), further supporting that the scaffold hybridized staples as intended (Supplementary Fig. 4b). Having determined biochemically that these RNA-scaffolded tetrahedra folded compactly, we next characterized their tertiary structure with cryo-electron microscopy (cryo-EM). Cryo-EM micrographs for the rT66 (Fig. 2c and Supplementary Fig. 5) showed monodisperse tetrahedral particles, with the reconstruction at 12 Å resolution showing slightly bowed edges, correlating overall with the target atomic model with a correlation coefficient of 0.76.

Turning to a more complex geometry with twelve edges of equal length and six four-way vertices, we used a 1056-nt transcript of the M13 phage genome to scaffold a regular octahedron with four helical turns per edge (rO44). The M13 transcript-scaffolded rO44 likewise formed a discrete shifted gel band (Fig. 2d), with cryo-EM screening showing monodisperse octahedral particles (Fig. 2e and Supplementary Fig. 6). Reconstruction from the cryo-EM data achieved a resolution of 17 Å, with the map showing a 0.90 correlation with the predicted atomic model. Unlike the rT66, bowed edges were not evident in the reconstructed rO44, possibly due to shorter edge lengths endowing relatively higher bending rigidity.

The rT66 and the rO44 reconstructions showed edge lengths corresponding to an average helical rise of 0.286 and 0.291 nm/bp, respectively (approximately 12.8 nm per edge for the rO44 and 18.9 nm per edge for the rT66), which are ~10–11% larger than the canonical A-form rise of 0.260 nm/bp and ~7–8% larger than theoretical rise of 0.267 nm/bp[67] from energy-minimized A-form duplex simulations. Although still generally consistent with expectations for A-form helices, this increase in average rise might indicate that the helices were slightly underwound[67] or that the crossover junction geometry created space that modestly lengthened the edges. For both the rT66 and rO44, the two duplexes in the edges moderately twisted relative to one another (Fig. 2c, e). This distortion, which has also been observed in molecular dynamics simulations[68], might indicate an inability of the A-form twist to relax fully in these structures.

Having validated folding of A-form DX wireframe origami with smaller geometries and scaffolds, we moved to the larger, natively structured and naturally abundant scaffold, ribosomal RNA (rRNA). We tested the ability of an in vitro-transcribed 1980-nt fragment of the *E. Coli* 23S rRNA to scaffold two different A-form DX wireframe origami objects of varying complexity: a regular octahedron and a pentagonal bipyramid, each with six helical turns (66 bp) per edge (rO66 and rPB66, respectively), as well as a pentagonal bipyramid with five helical turns (55 bp) per edge (rPB55). Unlike the other geometries investigated, the pentagonal bipyramid has multiple vertex types, with both four-way and five-way vertices and corresponding variations in dihedral angles, making it a more complex target geometry in addition to using the longer, intrinsically structured ribosomal RNA scaffold. Gel mobility shift assays suggested each object folded properly (Fig. 3a, d and Supplementary Fig. 7), with additional biochemical evidence provided by DMS-MaPseq showing normalized DMS reactivities among double helical segments lower in folded origami than in the 23S fragment scaffold folded without staples (72% lower for rO66, 71% lower for rPB66) (Fig. 3b, e). In further support of specific staple hybridization, the rO66 design only has staples hybridized to the first 1584 nucleotides of the scaffold, and we determined that the DMS reactivities in the excess scaffold region were well-correlated with those in the same region for the scaffold folded without staples (Pearson Correlation Coefficient=0.89), rather than being suppressed (Supplementary Fig. 8).

Cryo-EM micrographs for each particle also showed well-folded, monodisperse particles (Fig. 3c, f and Supplementary Figs. 9 and 10), with the rO66 reconstruction achieving 13 Å resolution with a correlation of 0.85 with the target, predicted atomic model. Reconstruction of the rPB66 achieved 19 Å resolution, with a correlation of 0.92

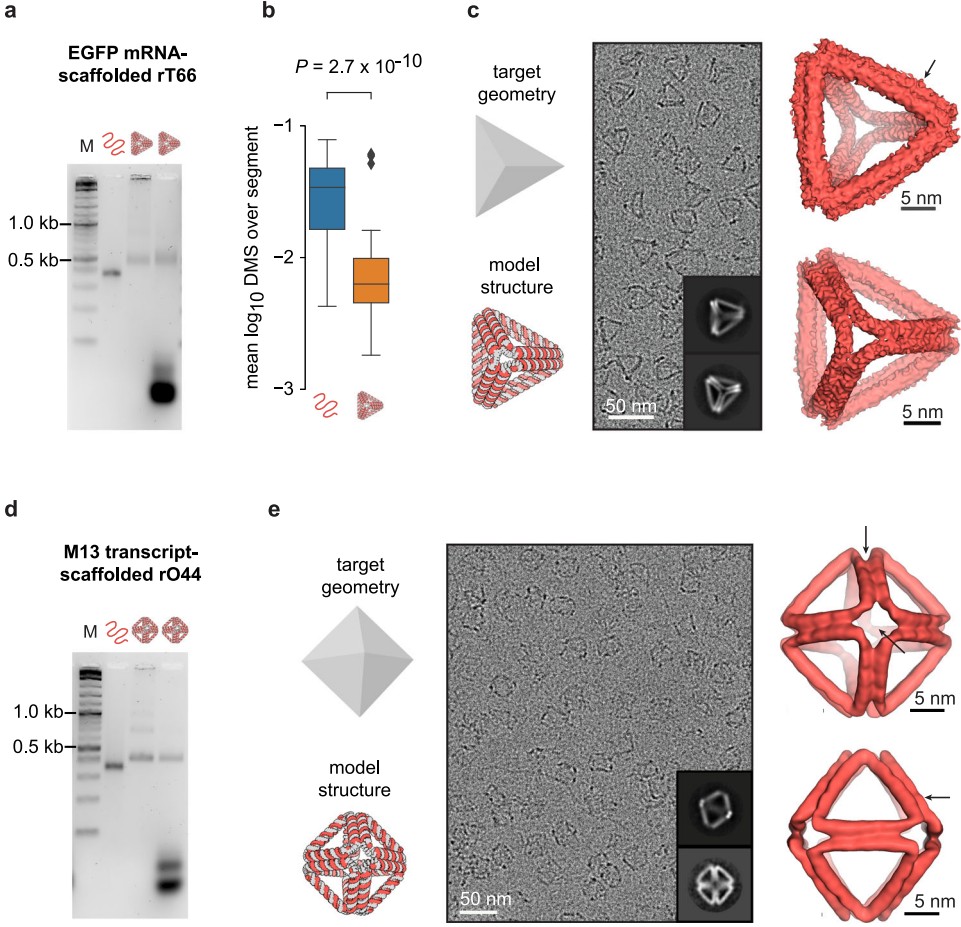

**Fig. 2 | Characterization of EGFP mRNA- and M13 RNA-scaffolded origami. a** Gel mobility shift assay, left to right: marker (1 kb plus DNA ladder, NEB), EGFP mRNA scaffold without DNA staples, spin filter-purified folded rT66, and unpurified folded rT66, $n = 1$ replicate. No scaffold band appears in the folded particle lane, suggesting 100% folding yield. **b** Box plots showing, for each unbroken segment of RNA:DNA duplex in rT66 ($n = 64$ segments), the mean DMS reactivity among adenines and cytosines in the segment's RNA scaffold strand (right, orange) compared with the same adenines and cytosines in the RNA strand folded without DNA staples (left, blue), $n = 1$ replicate. Each box plot depicts the median (middle line), 1st and 3rd quartiles (box), minimum/maximum up to 1.5 interquartile ranges from box (whiskers), and outliers (grey diamonds). The $P$-value indicates the significance of

the difference between the left and right distributions (two-sided Wilcoxon signed-rank test). **c** The input target geometry and predicted DX wireframe atomic model for the rT66, followed by an example micrograph (scale bar: 50 nm), two 2D class averages (insets), and two views of the reconstructed density map (scale bars: 5 nm), $n = 1$ replicate. Arrow indicates modest edge bowing. **d** Gel mobility shift assay, left to right: marker (1 kb plus DNA ladder, NEB), M13 transcript scaffold without DNA staples, spin filter-purified folded rO44, and unpurified folded rO44, $n = 1$ replicate. **e** The input target geometry and predicted DX wireframe atomic model for the rO44, followed by an example micrograph (scale bar: 50 nm), two 2D class averages (insets), and two views of the reconstructed density map (scale bars: 5 nm), $n = 1$ replicate. Source data are provided as a Source Data file.

between density map and predicted atomic model. In the cryo-EM reconstructions for both objects, we again observed a slight twisting of the two duplexes constituting the edges. The rO66 edges additionally exhibited outward bowing, which is not apparent in the rPB66 density map. Edges of the rO66 were approximately 17.9 nm long, corresponding to an average rise of 0.271 nm/bp. The rPB66 edges averaged approximately 17.8 nm long, corresponding to an average rise of 0.269 nm/bp. These values are consistent with the canonical A-form rise of 0.260 nm/bp and the theoretical A-form rise of 0.267 nm/bp from energy-minimized duplex simulations[67]. In both the rO66 and rPB66 reconstructions, we observed an offset in the duplex ends in each DX edge, likely corresponding to the pitch of A-form helices and the asymmetry in staple crossover designs.

As opposed to the A-form design we implemented for the origami described above, with 11 nt per helical turn and asymmetry in the staple crossover positions, previously reported RNA:DNA origami assemblies used different routing schemes[57], and optimal implementation of crossover asymmetry was not obvious for DX wireframe polyhedral origami. In contrast to single-stranded RNA tiles[54] and 2D origami bundles or tiles[57], DX designs for wireframe polyhedra are constrained

by the helical position of the scaffold exiting and entering duplexes at vertices, which has the potential to affect optimal design of crossover asymmetry. We, therefore, tested several routing scheme variants for our DX wireframe origami, including (1) a negative control B-form design with no crossover asymmetry and 10.5 bp per helical turn, (2) a symmetric A-form (Sym A-form) design with no crossover asymmetry and 11 nt per helical turn, and (3) an alternative A-form (Alt A-form) design with asymmetry incorporated into scaffold crossovers and 11 nt per helical turn (Supplementary Fig. 11). The latter design maintained the asymmetrical spacing between adjacent scaffold and staple crossovers on neighboring helices, which was proposed by models of A-form DX routing in RNA origami tiles[63], but in this case incorporating asymmetry into the scaffold instead of the staple crossover positions.

Folding the EGFP mRNA scaffold using staples designed for a B-form fold showed a notably higher gel mobility shift compared with the A-form fold, and cryo-EM micrographs did not show folded tetrahedral particles (Supplementary Fig. 12), as expected given that RNA is not known to adopt B-form geometry. On the other hand, the Sym A-form rT66 showed high folding yield, with cryo-EM micrographs showing well-formed tetrahedral particles, and a reconstruction

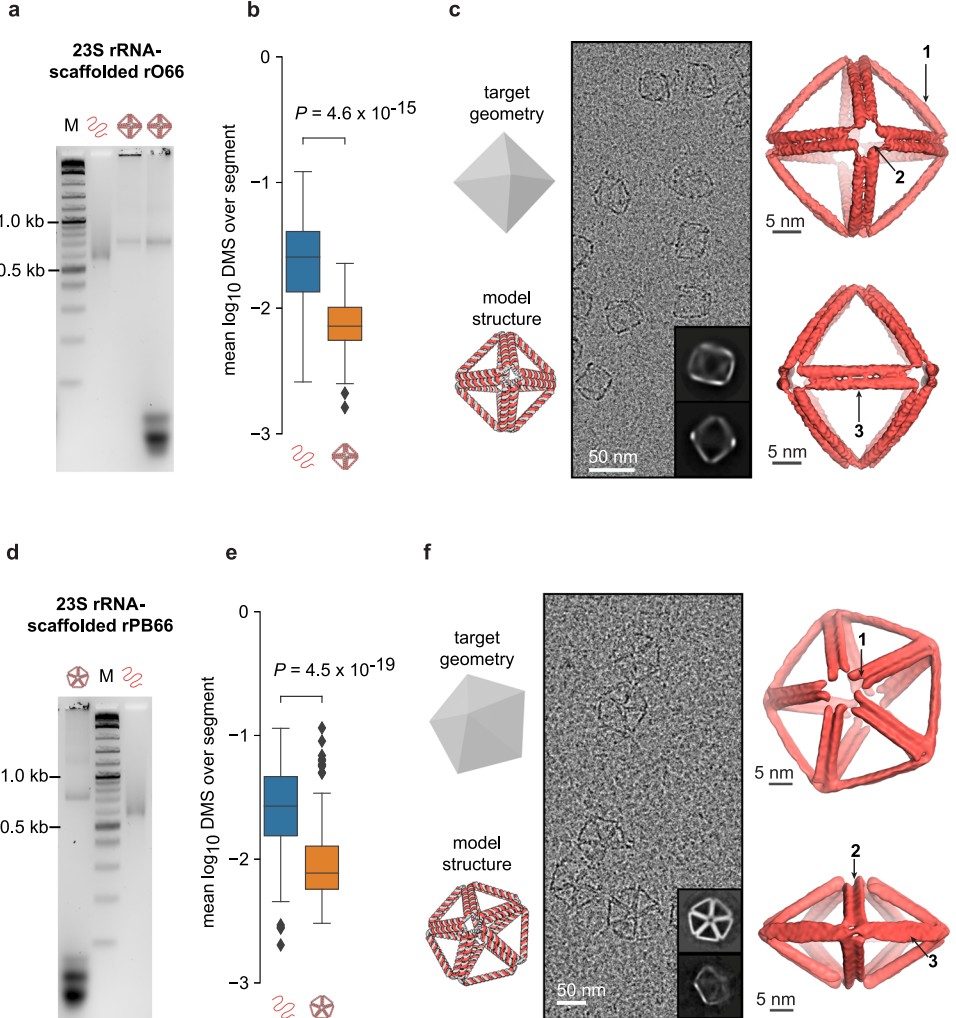

**Fig. 3 | Characterization of 23S rRNA-scaffolded origami. a** Gel mobility shift assay, left to right: marker (1 kb plus DNA ladder, NEB), 23S rRNA fragment scaffold without DNA staples, spin filter-purified folded rO66, and unpurified folded rO66, *n* = 1 replicate. **b** Box plots showing, for each unbroken segment of RNA:DNA duplex in rO66 (*n* = 134 segments) the mean DMS reactivity among adenines and cytosines in the segment's RNA scaffold strand (right, orange) compared with the same adenines and cytosines in the RNA strand folded without DNA staples (left, blue), *n* = 1 replicate each. Each box plot depicts the median (middle line), 1st and 3rd quartiles (box), minimum/maximum up to 1.5 interquartile ranges from box (whiskers), and outliers (grey diamonds). Each *P*-value indicates the significance of the difference between the left and right distributions (two-sided Wilcoxon signed-rank test). **c** The input target geometry and predicted DX wireframe atomic model

for the rO66, followed by an example micrograph, two 2D class averages (insets), and two views of the reconstructed density map, *n* = 1 replicate. Arrows indicate 1. Slight bowing in the DX edge, 2. Apparent twist in the vertex due to offset helical ends, and 3. Twist in the DX edge. **d** Gel mobility shift assay, left to right: unpurified folded rPB66, marker (1 kb plus DNA ladder, NEB), 23S rRNA fragment scaffold without DNA staples, *n* = 1 replicate. **e** Box plots of mean DMS reactivities as in panel **c**, but for the rPB66 (*n* = 166 segments). **f** The input target geometry and predicted DX wireframe atomic model for the rPB66, followed by an example micrograph, two 2D class averages (insets), and two views of the reconstructed density map, *n* = 1 replicate. Arrows indicate 1. The offset in the ends of helices due to the A-form pitch, 2-3. Indications of twist in the DX edges. Source data are provided as a Source Data file.

yielding a density map with 0.96 alignment correlation with the predicted model (Supplementary Fig. 13). However, the rO44 folded poorly with Sym A-form staples, yielding a considerably greater gel mobility shift than the A-form staples, and octahedral particles were absent in cryo-EM micrographs (Supplementary Fig. 14).

The Alt A-form rT66 and rPB66 designed with scaffold crossover asymmetry behaved similarly to their staple asymmetry A-form counterparts above. Salt titrations for folding the Alt A-form rT66 confirmed that 300 mM monovalent salt and 10 mM HEPES-KOH pH 7.5 was an appropriate buffer choice for folding these particles, despite their alternative routing scheme (Supplementary Figs. 15–17). Enzymatic digestion of the Alt A-form rT66 confirmed the structure was indeed RNA-scaffolded, and trace template DNA was not responsible for forming the principal origami product. RNase H, which specifically targets RNA hybridized to DNA, degraded the folded origami and

released staples within five minutes, with the folded origami band completely disappearing, the scaffold strand appearing to be fully digested, and bands corresponding to DNA staples concomitantly appearing (Supplementary Fig. 18). Dynamic light scattering (DLS) of the Alt A-form rT66 showed primarily monomeric populations with a peak at the hydrodynamic diameter 17.86 nm ± 5.67 nm and 34% polydispersity on average (Supplementary Fig. 19), similar to the 34% polydispersity observed for an analogous DNA-scaffolded DX-based tetrahedron with six helical turns (63 bp) per edge assuming B-form duplex geometry (Supplementary Fig. 20). The Alt A-form rT66 and rPB66 additionally showed well-folded particles in cryo-EM imaging (Supplementary Figs. 13, 21–22).

Taken together, these foregoing results suggest that RNA:DNA origami structures tolerate a range of routing designs that use 11 nt per helical turn, although inclusion of crossover asymmetry is important

to folding some geometries, like the octahedron. Based on inspection of predicted atomic structures that assume an ideal helical position of scaffold exiting duplexes at vertices, the proposed A-form design better minimizes strain and steric clash at crossover positions than the Alt A-form design (Supplementary Figs. 23 and 24).

The A-form design for DX wireframe origami was incorporated as the A-form option in the open-source pyDAEDALUS software, which generates scaffold and staple routing for polyhedral mesh surfaces to design DX-edge-based wireframe structures (Fig. 1). 14 polyhedra designed by the software with A-form scaffold and staple routing resulted in well-matched predicted structures compared with target input geometries for a variety of scaffold sequences (Supplementary Fig. 25), including mRNA sequences that may eventually be of interest for applications in mRNA delivery (GenBank[69] accession numbers NM_002111.8 and M28668.1), the mRNA sequence used for SARS-CoV-2 vaccination[70], a long replicon RNA followed by mCherry coding sequence[71], ribosomal RNA[72], and viral genomes (GenBank[69] accession numbers GU131973.1 and KY008770.1) that could serve as scalable sources of long single-stranded RNA to fold larger objects for nanomaterials applications.

## Base-pair-level insight into origami secondary structure stability using DMS-MaPseq

To gain base-level insight into how the placement of crossovers, strand termini, and vertex designs impact RNA-scaffolded origami stability, we analyzed DMS-MaPseq data for nucleotides near these features across the origami (Fig. 4a). For each type of structural feature, we identified all instances of the structural feature among $n = 5$ characterized A-form origami objects (rT55, rT66, rT77, rO66, and rPB66) and computed the distributions of DMS reactivities at each position from 6 bp upstream to 6 bp downstream of the feature instance, where 6 bp corresponded to the shortest contiguous double helical segment in the origami objects investigated (Fig. 4b and Methods). The bases immediately upstream (−1) and downstream (+1) of staple nicks, staple single termini, and vertices were more DMS-reactive than the corresponding bases further upstream or downstream (positions −6 to −2 and +2 to +6, respectively) at a significance level of 0.01 (one-tailed Mann Whitney $U$ test). The same was true for bases immediately upstream of staple double and single crossovers. However, by the same criteria for significance, none of these more distant bases was more reactive than corresponding bases further away from the structural feature, except for bases two positions upstream of vertices ($P = 5 \times 10^{-5}$, one-tailed Mann Whitney $U$ test). Thus, crossovers and staple termini destabilized only the immediately adjacent bases, and vertices only destabilized bases within two positions.

To account for any biases caused by differences in innate DMS reactivities between adenines and cytosines, we repeated the analysis on each type of residue separately (Supplementary Fig. 26). In each case, bases further than 1 nt from a structural feature did not have elevated DMS reactivities, except for bases 2 nt upstream of vertices. For adenines alone, the reactivities of bases adjacent to structural features were generally larger and the $P$-values generally lower than for cytosines alone. This result suggested that stabilities of A-T base pairs depended more on their proximities to structural features than do C-G base pairs, as might be anticipated due to their weaker hybridization free energy.

Uniquely among structural features, scaffold double crossovers showed elevated DMS reactivities at all 6 positions upstream, although bases immediately upstream (position −1) were no more reactive than those further upstream (Fig. 4b). All scaffold double crossovers were placed downstream of a staple nick by 6 or 7 bp—the shortest distance between two structural features in the origami. We predicted the melting temperature of each segment in every origami (Methods) and found that the shortest segments tended to have the lowest predicted melting temperatures (Fig. 4c, light blue circles). Thus, we inferred that

the low melting temperatures of these short double helical segments near scaffold crossovers reduced their tendencies to hybridize, as supported by previous DNA nanotechnology sequence designs[73,74]. In further support of this interpretation, amongst all contiguous double stranded segments in the origamis, mean DMS reactivities among the interior bases (excluding the 5′ and 3′ ends of the segment) were moderately anticorrelated with predicted melting temperature ($\rho = -0.58$)(Fig. 4c), as well as with segment length ($\rho = -0.41$) and GC content ($\rho = -0.41$)(Supplementary Fig. 27). Furthermore, in longer segments with staple nicks at their 5′ ends, interior bases had low DMS reactivities (Fig. 4d), suggesting that the 5′ staple nick was not the main cause of the high DMS reactivities in the 6 or 7 bp segments. We could not deconvolute the contributions of 3′ scaffold crossovers versus short segment lengths, but given that the effects of all other structural features were limited to a one- or two-nucleotide vicinity (Fig. 4b) and that segments with 5′ scaffold crossovers generally had low DMS reactivities, 3′ scaffold crossovers were unlikely to be the primary cause of the increased DMS reactivities of interior nucleotides.

To further investigate the relative stabilities of different structural features, we divided the DMS reactivity of the base immediately downstream (or upstream) of the structural feature by the mean DMS reactivity among the interior bases in the adjacent segment, isolating the effects of each structural feature on local stability when located at the 5′ (or 3′) end of a segment, while controlling for factors that could affect the entire segment (e.g., melting temperature)(Fig. 4d). Bases adjacent to staple single crossovers, staple termini, and vertices, as well as immediately upstream of staple double crossovers, all tended to be more DMS-reactive than the interiors of the adjacent segments at a significance level of 0.05 (one-tailed Wilcoxon signed-rank test). Staple termini (both nicks and single termini) and vertices all destabilized local hybridization approximately equally, and more than staple and scaffold double crossovers alone (Fig. 4d). DMS reactivities (relative to interior bases) were greater for bases immediately upstream and downstream of staple single crossovers than double crossovers ($P = 2 \times 10^{-5}$ and $P = 1 \times 10^{-2}$, respectively, two-tailed Mann-Whitney $U$ test). These findings are consistent with previous work with DNA origami motifs that showed mesojunctions, the geometry adopted by single crossovers in DX edges, are less thermally stable than the conventional junctions of double crossover motifs[75,76].

To glean insights for optimizing sequences of RNA scaffolds for folding polyhedral wireframe origami, we investigated differences between adenines and cytosines adjacent to each structural feature (Fig. 4e). Most notably, for segments with a 3′ staple single crossover, adenine residues at the crossover were nearly four times as DMS reactive as cytosine residues ($P = 1 \times 10^{-3}$, two-tailed Mann–Whitney $U$ test), relative to interior adenines and cytosines, respectively. Similarly, terminal adenines (relative to interior adenines) were more reactive than terminal cytosines (relative to interior cytosines) for segments with 3′ staple single termini and staple nicks. This finding suggests that A-T pairs immediately upstream of staple single crossovers, single termini, and nicks are particularly unstable, and that these sites in origami might be most important to stabilize with C-G pairs. Conversely, terminal cytosines were approximately two-fold more reactive than terminal adenines (relative to interior residues of the same type) for segments with 5′ staple nicks and 5′ staple double crossovers. Relative to interior bases of the same type, there were no significant differences in the reactivities of adenines versus cytosines adjacent to scaffold double crossovers and vertices. These results show that, after controlling for the type of residue and DMS reactivity of each segment, structural features destabilize adjacent A-T and C-G base pairs to different extents, which has implications for optimizing the sequence of the scaffold. For example, prioritizing placement of cytosines and guanines at and near structural features, when feasible based on sequence design and fabrication constraints, should enhance origami stability. While these base-level findings are limited here to

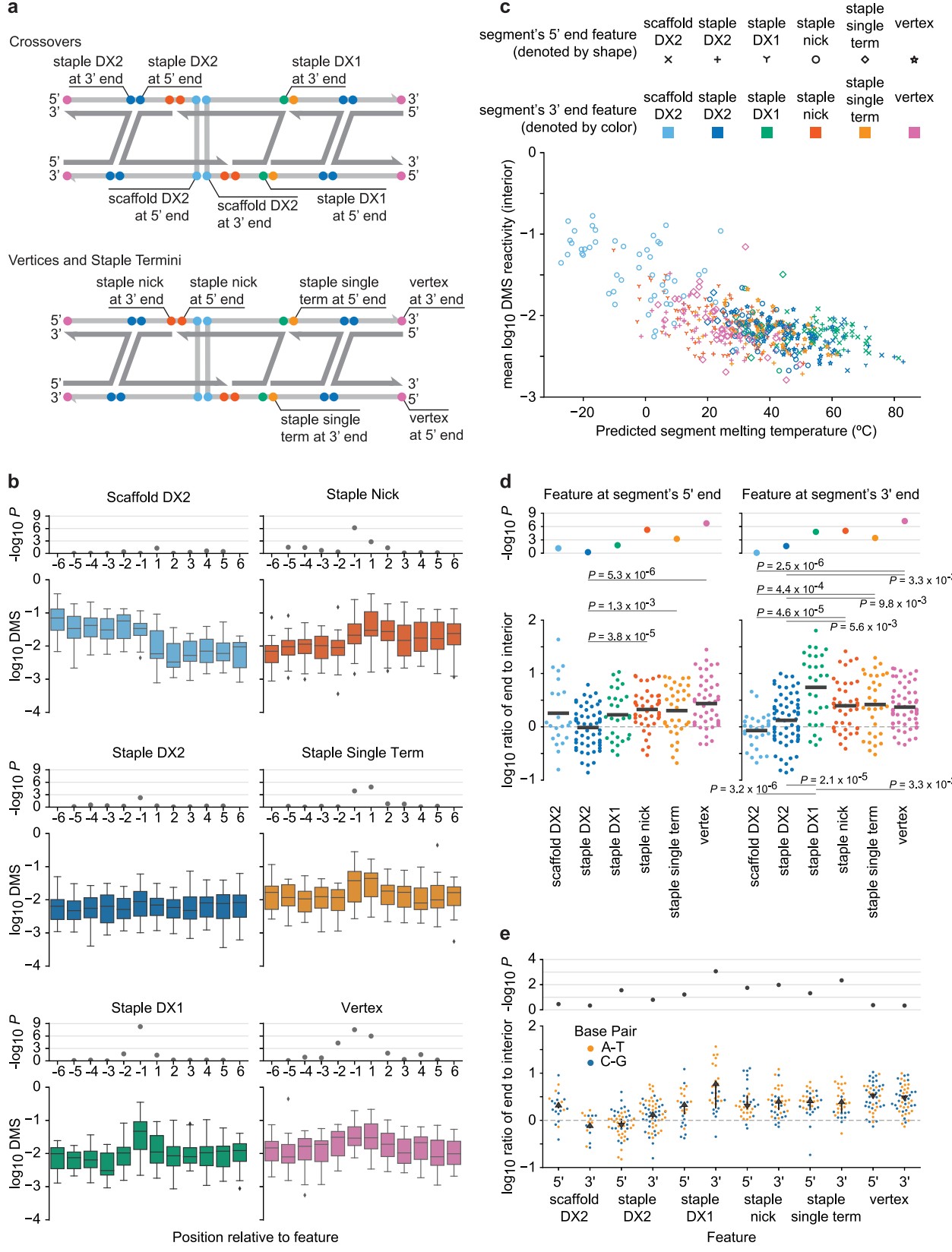

RNA:DNA origami, they are likely to generalize to DNA:DNA origami as well.

## Discussion

We demonstrated folding of a variety of RNA scaffolds using DNA staples to self-assemble, via thermal annealing, distinct 3D polyhedral wireframe geometries based on DX-edge designs. Aside from the need to accommodate A-form helical geometry, a major concern in using long RNA strands to scaffold origami has been its tendency to be highly structured internally, as observed in long natural RNAs from rRNA to viral genomes[66,77,78], as well as the relative unpredictability of its tertiary structures. However, the presence of native secondary and

**Fig. 4 | Per-nucleotide stabilities of RNA-scaffolded origami revealed by DMS-MaPseq. a** Dual-duplex antiparallel crossover (DX) edge comprising RNA scaffold (light grey) and DNA staple (dark grey) strands. One example of each structural feature is labeled on the involved scaffold base (because DMS-MaPseq probes only RNA). DX1: single crossover, DX2: double crossover, term: terminus, 5′ and 3′: ends (with respect to scaffold) of unbroken segments of RNA:DNA duplex.
**b** Distributions of DMS reactivities among adenines and cytosines within 6 bp of each type of feature in $n = 5$ independent A-form origami (rT55, rT66, rT77, rO66, rPB66). Positions −1 and 1, respectively, lie immediately 5′ and 3′ of the feature and correspond to segment 3′ and 5′ ends labeled in (**a**). Each box plot depicts the median (middle line), 1st and 3rd quartiles (box), minimum/maximum up to 1.5 interquartile ranges from box (whiskers), and outliers (grey diamonds). *P*-values indicate significance of median reactivities exceeding those among all positions further from the feature on the same side (one-sided Mann-Whitney *U* test).
**c** Comparison of predicted melting temperatures and mean DMS reactivities

among interior (excluding 5′ and 3′ ends) adenines and cytosines of $n = 490$ segments. Each point depicts one segment whose 5′ and 3′ end features determine the point's shape and color, respectively. **d** Destabilization of base pairs at segment ends. For each segment, the ratio between the DMS reactivity of its 5′ (3′) terminal nucleotide (if adenine or cytosine) and the mean DMS reactivity among its interior adenines and cytosines is plotted on the left (right), above the feature at the 5′ (3′) end of the segment. Black bars show means. Upper *P*-values indicate significance of the median ratios exceeding unity (one-tailed Wilcoxon signed-rank test); *P*-values connecting pairs of features indicate significance of differences between the median ratios of the features (two-tailed Mann–Whitney *U* test). **e** Similar to (**d**), but separately analyzing DMS reactivities of adenines (A-T pairs) and cytosines (C-G pairs). Arrows point from mean ratios among C-G pairs to A-T pairs. *P*-values indicate significance of differences between the median ratios of A-T and C-G pairs (two-tailed Mann–Whitney *U* test). Source data are provided as a Source Data file.

tertiary structure present in target RNA scaffolds did not appear to prevent proper folding, given that the 23S rRNA fragment used to scaffold three different objects has a variety of secondary structural motifs and is approximately ~58% base paired on its own according to our DMS-MaPseq analysis. Following denaturing at high temperatures and re-annealing, this RNA scaffold bound preferentially to the staples and folded the target structure with high yield. The assumption of A-form helices in our sequence design algorithm may also apply to RNA staples with DNA scaffolds, or entirely RNA-RNA wireframe origami.

Scaffolding wireframe origami with RNA allowed us to study nucleic acid origami folding and stability with nucleotide-level precision. Further, although numerous studies have investigated the stability of and defects in DNA origami at the staple level, for example using fluorescent probes to estimate the average number of unpaired nucleotides in 3D helix bundles[39] and high-resolution atomic force microscopy to visualize missing staples directly in 2D tiles[40], chemical probing to characterize base-level secondary structure has only been applied to simple 3-way and 4-way DNA junctions using hydroxyl radical footprinting[76]. The investigation of large-scale scaffolded origami using chemical probing was made possible in this work by the use of RNA scaffolds. Our observations of secondary structure stability might also generalize to scaffolded DNA origami. We can apply this protocol to investigate base pairing stability for improved sequence design—for example, our data suggest that placing C-G base pairs immediately upstream of staple single crossovers should increase origami stability. Complementing existing techniques[79–81], the approach also holds promise for kinetic studies of nucleic acid origami folding, by probing samples either at various time points of isothermal co-transcriptional folding[53,55] or at various stages of thermal annealing and comparing which sections of the scaffold are hybridized at each stage, which may improve mechanistic understanding of scaffolded origami folding.

Beyond studies of nucleic acid origami itself, the use of RNA as a scaffold for DX wireframe origami enables a variety of applications, depending on the particular scaffold used. RNA has a number of useful features that are distinct from DNA, including the ability to modify nucleotides for stability, immunogenicity, and translatability, and to introduce riboswitches and aptamers, ribozymes, antisense oligos, and long RNAs. In particular, here we used mRNA as a scaffold for the assembly of a nanoparticle containing the sequence encoding a fluorescent protein. Targeted cellular delivery of such a nanoparticle could offer important potential for nuclease-specific release of scaffold or staples within the cell, with applications in antisense oligonucleotide (ASO) therapy[82–84], multiplex automated genome engineering (MAGE)[85], and homologous recombination template[86] delivery. The tunable degradation rates introduced by the combined use of RNA, modified RNA and DNA may also prove useful for material templating and etching. We additionally used ribosomal RNA to scaffold a

pentagonal bipyramid that would leave domains V and VI of the 23S rRNA to fold freely. Future engineering might allow for the generation of synthetic nucleic acid assemblies that coordinate catalytic ribozymes[87,88], test substrates of RNA modification enzymes[89,90], and eventually even mimic or augment the functions of the ribosome during protein synthesis[91–93].

## Methods

### Reagents
Oligonucleotide staples and primers and gBlock synthetic DNA sequences were purchased from Integrated DNA Technologies (IDT, Coralville, IA). HEPES, Trizma base, EDTA, NaCl, KCl, MgCl$_2$, magnesium acetate, acetic acid, and high-resolution agarose were purchased from Sigma-Millipore (MA). HiScribe T7 RNA polymerase kits and Q5 2x HiFidelity PCR mastermixes were purchased from New England Biolabs (NEB, Ipswich, MA). Sodium cacodylate solution, pH 7.2, was purchased from VWR.

### A-form DX wireframe origami programmed using pyDAEDALUS
To generalize to A-form helical geometries to allow for RNA:RNA or hybrid RNA:DNA DX-based origami, edge lengths were discretized to multiples of 11 rather than rounded multiples of 10.5, and crossover positions were changed to be compatible with A-form helix crossovers. As described previously[54], scaffold crossover edges, which have adjacent crossovers occurring on different strands of the double helix (scaffold vs. staple) and thus must occur an odd number of half-twists apart, require that the crossovers be spaced asymmetrically on the two duplexes of the edge to be compatible with A-form helical geometry. We investigated two approaches to implementing this asymmetry (A-form and Alt A-form), in addition to testing a design with no asymmetry incorporated (Sym A-form).

In one approach to A-form, the required asymmetry is incorporated into the staple crossover position calculation (Fig. 1, Supplementary Fig. 11). In this case, staple crossovers are asymmetric across the two duplexes, with a 4-nt difference between the nucleotide position on the two duplexes (e.g., the first staple crossover occurs 9 nt from the vertex on the 5′ side and 13 nt from the vertex on the 3′ side).

An alternative approach (Alt A-form) incorporates asymmetry into the scaffold crossover position calculation for A-form (Supplementary Fig. 11). In this approach, the scaffold crossover has a 5-nt difference between the nucleotide position on the two duplexes (e.g. the longer scaffold crossover half on a 44-nt edge would occur 25 nt from the vertex on the 5′ side and 30 nt from the vertex on the returning 3′ side). Manual modifications from B-form designs to the Alt A-form were initially implemented using Tiamat software[94] and subsequently automated. For the tetrahedron and octahedron, a Sym A-form design was also manually generated, which does not have the asymmetry in crossover position between the two duplexes, but instead directly crosses over as in the standard DNA DX design (Supplementary Fig. 11).

We implemented the A-form design rules with staple crossover asymmetry in a top-down design algorithm that calculates sequences for folding an input target shape with wireframe DX edges (Fig. 1). The code architecture in pyDAEDALUS (https://github.com/lcbb/pyDAEDALUS) mirrors the format and naming conventions of DAEDALUS[18]. Briefly, the input target geometry file in the Polygon File Format (PLY) is parsed to identify relevant geometric parameters including coordinates of vertices, edge and face connectivities that form the graph of the shape, and edge lengths. Scaffold routing is achieved by calculating the spanning tree of the graph, and staples are added according to either standard geometric rules for B-form DNA or the A-form design rules, depending on user specification. The resulting outputs of the algorithm are plaintext and Comma-Separated Values (CSV) files that store the routing information, staple sequences, and nucleotide spatial coordinates. The positions and orientations of each nucleotide are represented as vectors following the convention from the software 3DNA[95].

While the overall architecture of DAEDALUS is preserved in pyDAEDALUS, several fundamental changes were required. First, the connectivities passed through the functions are stored in pyDAEDALUS as NetworkX Graph objects, rather than sparse matrices as in DAEDALUS. Second, Prim's algorithm[96], which is used to generate the spanning trees required to route the scaffold strand, was used in both algorithms as built-in functions. However, for many structures the Python version generates a spanning tree different from the MATLAB version. Although this will affect the scaffold routing and staple sequences, the fidelity of the final design should not be affected, because each possible spanning tree of an object corresponds to a valid scaffold routing[18]. Third, in order to exploit the object-oriented structure that Python enables, the DNAInfo class was introduced, which packages together the many variables associated with the geometry, routing, and structure generated in intermediate sub-functions of the algorithm.

To render the code more robust and offer a platform for further development by other contributors, additional frameworks were constructed. A style guide was implemented to help readability of the code, and linting, i.e. automatic checking of adherence to the style guide, is also enforced. In addition, unit tests were introduced to ensure that the functionality of the code is preserved as intended by the original authors. All tests and linting are evaluated automatically on each change to the current version on GitHub, with the results published within the readme.md of the repository.

## RNA transcription

The full sequence of each DNA template is listed in Supplementary Data 1. For the RNA-scaffolded tetrahedron, the EGFP sequence was generated as a gBlock and cloned with a T7 promoter and Shine-Dalgarno (SD) sequence 5′ of the coding sequence into a pUC19 vector using restriction cloning (EcoRI, PstI). RNA was transcribed from a Phusion PCR-generated double-stranded DNA (dsDNA) template containing a 5′ T7 promoter, amplified using primers listed in Supplementary Data 2, and gel purified. For the pentagonal bipyramid and octahedron with 6 helical turns per edge, primers were chosen flanking Domains I–IV of the *rrlB* gene encoding the 23S rRNA from the pCW1 plasmid[72]. For the octahedron with 4 helical turns per edge, partial M13 DNA template was amplified from mp18 ssDNA (NEB). For the fragment of human immunodeficiency virus 1 (HIV-1) Rev response element (RRE) used as a control in DMS-MaPseq experiments, the sequence was synthesized as a gBlock (IDT) containing a 5′ T7 promoter, then PCR amplified using the Q5 High-Fidelity 2X Master Mix (NEB). For the tetrahedra with 5 and 7 helical turns per edge, we designed a random scaffold sequence with minimal self-complementarity (rsc1218v1, see 'De Bruijn scaffold' in Methods) and again obtained the sequence as a synthetic gBlock (IDT), then amplified with Q5 High-Fidelity 2x Master Mix (NEB) to create dsDNA templates for 660 nt and 924 nt scaffolds, each with a 5′ T7 promoter.

Using these dsDNA templates, RNA was transcribed using the manufacturers protocol for HiScribe T7 (NEB). RNA was treated with DNase I (NEB), then pre-cleaned on a ZymoClean RNA cleanup kit (RNA Clean-and-concentrator 5). Urea polyacrylamide gel (PAGE) was used to validate purity, and PAGE or HPLC was used to purify the RNAs if byproducts were present. With RNA pre-cleaned using the RNA clean-and-concentrator-5 kit (Zymo), and denatured by addition of 1x RNA loading dye (NEB) and 5–10 min incubation at 70 °C, PAGE purification was performed on a 6% gel containing 8 M urea. RNA was sliced from the gel after visualization with SYBR Safe (ThermoFisher) and eluted in 300 mM sodium acetate pH 5.2, precipitated in 70% ethanol at −20 °C for >2 h, and then pelleted at 14,000 RPM for 30 min at 4 °C.

For HPLC purification, transcribed and column-purified (with ZymoClean RNA clean-and-concentrator-5 kit) RNA was diluted with nuclease-free water and injected into an XBridge Oligonucleotide BEH C18 column (130 Å, 2.5 μm, 4.6 mm × 50 mm, Waters) under the following gradient, flowing at 0.9 ml/min: increasing from 38–40% solvent B over 1 min, increasing to 60% buffer B across 15 min, increasing to 66% buffer B across 6 min, increasing to 70% buffer B across 30 s, reaching 100% buffer B across 30 s, maintaining 100% buffer B for 1 min, decreasing to 38% solvent B over 1 min, where it was finally held for 2 min (adapted from a previously-published protocol[97]). Buffer A was a solution containing 0.1 M TEAA, while buffer B included 0.1 M TEAA and 25% (v/v) acetonitrile. All HPLC purification of the RNA scaffold was run at 65 °C to prevent formation of secondary structure. Sodium acetate, pH 5.2, was added to a final concentration of 300 mM in the collected fraction, and the RNA was precipitated in 70% ethanol at −20 °C for >2 h, then pelleted at 14,000 RPM for 30 min at 4 °C.

## Hybrid RNA:DNA nanoparticle folding and characterization

The sequence of every DNA staple used in this study is listed in Supplementary Data 3. Unless otherwise specified, RNA:DNA origami wireframe particles were folded using 20 nM of purified scaffold mixed with 400 nM individual staples in 10 mM HEPES-KOH pH 7.5 and 300 mM KCl, all RNase-free buffers and conditions, in volumes of 50 μl aliquots. Folding was performed using a modification of the previously published wireframe origami thermal annealing protocol[18] but with reduced incubation time at high temperatures and in the modified buffer mentioned to reduce RNA instability. Briefly, the folding protocol was 90 °C for 45 s; ramp 85 °C to 70 °C at 45 s/°C; ramp 70 °C to 29 °C at 15 m/°C; ramp 29 °C to 25 °C at 10 m/°C; 10 m at 37 °C; hold at 4 °C until purification. Folded particles were purified away from excess staples using Amicon Ultra 100 kDa 0.5 ml filter columns and buffer exchanged into the same buffer used for folding.

Variations on the folding protocol described above were tested with the Alt A-form origami: Using RNase-free buffers and conditions, 20 nM of purified scaffold was mixed with 400 nM individual staples and buffer and salt and brought to 50 μl aliquots for temperature ramping. 10 mM and 50 mM HEPES-KOH pH 7.5 and 50 mM Tris-HCl pH 8.1 were tested, and salt concentrations were tested in 10 mM HEPES-KOH such that the final concentrations of KCl and NaCl individually were 0, 100, 200, 300, 400, and 500 mM, and for $MgCl_2$: 0, 2, 4, 8, 12, and 16 mM.

We also tested two alternative published protocols with the A-form origami. Following the protocol from Wang et al.[61], we mixed 20 nM purified scaffold with 200 nM individual staples in 40 mM Tris pH 8.0 with 20 mM acetic acid, 2 mM EDTA, and 12.5 mM magnesium acetate, and annealed by incubation at 65 °C for 10 min, 50 °C for 10 min, 37 °C for 10 min, 25 °C for 10 min, ending with a hold at 4 °C. Following the protocol from Zhou et al.[59], we mixed 20 nM purified scaffold with 200 nM individual staples in 5 mM Tris pH 7.5 with 1 mM EDTA and 40 mM NaCl, annealed using an overnight folding ramp (Starting at 65 °C, hold for 1.5 min per cycle and cycle 349x, decreasing by 0.1 °C each cycle, then at 30 °C hold for 2 min per cycle and cycle 99x, again decreasing by 0.1 °C each cycle, ending with a hold at 4 °C).

Gel mobility shift assays were performed using 2.5% high-resolution agarose (Sigma-Millipore) in 1x TAE and 2 mM magnesium acetate. Folded origami samples were prepared for loading with 1x purple DNA loading dye with no SDS (NEB). RNA scaffolds without DNA staples were prepared with 1x RNA dye (NEB) and denatured at 70 °C for 10 min or 95 °C for 2 min, followed by incubation on ice for at least 2 min immediately prior to gel loading. Gels were run on a Bio-Rad gel system in a cold room (4 °C) at 65 V for 4 h, then imaged using a Typhoon FLA 7000 scanner (GE).

The size distribution of the origami nanoparticles was measured via DLS using a Zetasizer Nano ZSP (model ZEN5600, Malvern Instruments, UK). Purified nanoparticles were concentrated to 75 nM in 50 μl in 10 mM HEPES-KOH pH 7.5 and 300 mM KCl. The default procedure for DNA was used, only customizing the buffer to include 300 mM KCl. Three serial DLS measurements were performed on the same folded sample at 25 °C. The average nanoparticle diameter (nm) and polydispersity index (PdI) were computed using the associated Malvern software (Zetasizer Software v 7.12).

Biochemical stability in the presence of RNases A and H was also tested. 50 nM purified Alt A-form EGFP mRNA-scaffolded tetrahedron with 66-bp edge length was incubated for 5 min at 37 °C in the presence of buffer alone, 25 units of RNase H or 3.5 units of RNase A. Reactions were quenched at 4 °C and run at 65 V for 180 min on a high-resolution 2.5% agarose gel in 1X TBE with 2.5 mM Mg(OAc)$_2$, maintained at 4 °C on ice.

### Chemical probing of secondary structure with DMS-MaPseq

The DNA template for each scaffold and the HIV-1 RRE fragment was generated, amplified, and transcribed into RNA as described above. Three denaturing polyacrylamide gels with 6 M urea (2.4 ml 5X Tris-Borate-EDTA (TBE), 4.32 g urea, 1.2 ml 40% 19:1 acrylamide:bis-acrylamide, 120 μl ammonium persulfate, 12 μl TEMED, nuclease-free water to 12 ml) were pre-run at 160 volts for 30 min. The RNAs were denatured in 2X RNA Loading Dye (NEB) at 70 °C for 10 min, immediately placed on ice, and run on the gels at 160 volts for 60 min in 1X TBE in a Mini-PROTEAN Tetra Cell (Bio-Rad). The gels were stained in 1X TBE containing 1X SYBR Safe (Thermo Fisher) for 5 min, and each band of expected molecular weight was excised and transferred to a 0.5 ml tube at the bottom of which a hole had been punctured with a needle. Each 0.5 ml tube was placed in a 1.5 ml tube and spun at 16,000 × g for 60 sec to extrude the gel slice into the 1.5 ml tube. Each gel slice was covered with 400 μl of gel elution buffer (250 mM sodium acetate pH 5.2, 20 mM Tris-HCl pH 8.0, 1 mM EDTA pH 8.0, 0.25% w/v sodium dodecyl sulfate) and incubated in a thermomixer at 20 °C for 11 hr while shaking at 500 rpm. The slurries were decanted into Costar Spin-X Centrifuge Tube Filters (Corning) and spun at 16,000 × g for 1 min to remove gel particles. To each filtrate, 1 ml 100% ethanol was added, and the tubes were frozen at −80 °C for 1 hr. The tubes were then spun at 12,700 × g for 1 hr at 4 °C. The pellets were washed with 500 μl 75% ethanol at −20 °C and spun for another 10 min. The supernatants were removed and the tubes uncapped and placed on a 37 °C heat block to dry the pellets for 10–20 min. The pellets containing the RNA were resuspended in 10 μl nuclease-free water.

The gel-purified RNA scaffolds were used to fold nanoparticles in folding buffer (10 mM HEPES-KOH pH 7.5, 300 mM KCl) using 20 nM scaffold and 400 nM for each staple with the temperature steps described above. Each RNA scaffold was also folded using the same protocol but without adding staples. The 23S scaffold was folded in two tubes each containing 85 μl; rT66 without staple 10 was folded in one tube containing 70 μl; all other nanoparticles and scaffolds were folded in two tubes each containing 60 μl.

Folded nanoparticles/scaffolds were purified by five rounds of filtration through Amicon Ultra 100 kDa 0.5 ml filter columns. One Amicon filter for each of the 16 nanoparticle/scaffold samples was first spun at 2400 × g for 30 min at 4 °C with 500 μl of the buffer used to fold the origami. To each pre-spun filter, 350 μl of 300 mM sodium cacodylate pH 7.2 (Electron Microscopy Sciences) was added, followed by 50–150 μl of the pooled folding product of one nanoparticle/scaffold. The samples were spun at 850×g for 30 min at 4 °C, after which the filtrate was decanted and 450 μl sodium cacodylate added to the filter, and these steps were repeated for a total of five filtrations. The fifth filtration was run for 50 min, after which each filter (containing approximately 50 μl) was inverted into a clean collection tube and spun at 1500 × g for 1 min at 4 °C to collect the sample of nanoparticles/scaffold. A 10 μl aliquot of each sample was transferred to a 1.5 ml tube.

As a control for normalization of the DMS reactivities, 1.3 μg of gel-purified RRE RNA was denatured in 8 μl of RNase-free water at 95 °C for 60 s and immediately placed on ice for 60 s. The denatured RRE RNA was mixed with 612 μl of 300 mM sodium cacodylate pH 7.2 (Electron Microscopy Sciences) and incubated at 37 °C for 20 min to refold its structure. A 38.5 μl aliquot of refolded RRE RNA was added into each of 16 tubes containing 10 μl of one nanoparticle/scaffold. To each sample, 1.5 μl of neat dimethyl sulfate (DMS, MilliporeSigma) was added (50 μl total volume, 3% DMS v/v), stirred with a pipette tip, and incubated at 37 °C for 5 min in a thermomixer while shaking at 500 rpm. Each reaction was quenched by adding 30 μl neat beta-mercaptoethanol (MilliporeSigma). DMS-modified nucleic acids were purified using a Zymo RNA Clean and Concentrator-5 Kit (Zymo Research) and eluted in 10 μl RNase-free water.

For each RNA sample, 4 μl was reverse transcribed in a 20 μl reaction containing 1 μl pooled reverse primers (10 μM each), 1 μl TGIRT-III enzyme (Ingex), 4 μl 5X First Strand buffer (Invitrogen), 1 μl 10 mM dNTPs (Promega), 1 μl 0.1 M dithiothreitol (Invitrogen), 1 μl RNaseOUT (Invitrogen), and 7 μl nuclease-free water. The reactions were incubated at 57 °C in a thermocycler with the lid set to 60 °C for 90 min. The RNA templates were degraded by adding 1 μl of 4.0 M sodium hydroxide (MilliporeSigma) to each reaction and incubating at 95 °C for 3 min. Each cDNA was purified using a Zymo Oligo Clean and Concentrator-5 Kit (Zymo Research) and eluted in 10 μl nuclease-free water.

The cDNA from each of the 16 samples was amplified as a set of overlapping amplicons, each 250–556 bp (47 amplicons total), plus one amplicon spanning the entire RRE (16 amplicons total). For each amplicon, 1 μl purified cDNA was amplified with an Advantage HF 2 PCR kit (Takara) in a 25 μl reaction containing 0.5 μl forward primer (10 μM, IDT), 0.5 μl reverse primer (10 μM, IDT), 0.5 μl 50x Advantage-HF 2 Polymerase Mix, 2.5 μl 10x Advantage 2 PCR Buffer, 2.5 μl 10x HF dNTP Mix, and 17.5 μl nuclease-free water. The PCR entailed an initial denaturation step at 94 °C for 60 s, followed by 25 cycles of 94 °C for 30 sec, 60 °C for 30 s, and 68 °C for 60 sec, with a final extension at 68 °C for 60 s. All PCR products were validated using E-Gel EX-Gels with 2% Agarose (Thermo Fisher).

All 47 PCR products from nanoparticles/scaffolds and 5 RRE products were consolidated into 5 pools such that no two amplicons from the same RNA sequence were pooled together. Pools 1 – 4 contained 6 μl each of 10 PCR products; pool 5 contained 5 μl each of 12 PCR products. For each pool, 30 μl was mixed with 6 μl 6X gel loading dye and run on a 50 ml gel containing 2% SeaKem Agarose (Lonza), 1x Tris-Acetate-EDTA (Boston BioProducts), and 5 μl 10,000X SYBR Safe DNA Gel Stain (Thermo Fisher) at 60 volts for 105 min. Bands at the expected sizes were excised and the DNA was extracted using a Zymoclean Gel DNA Recovery Kit (Zymo Research) and eluted in 12 μl 10 mM Tris pH 8.0 (MilliporeSigma). DNA libraries were generated and sequenced on an Illumina MiSeq using a 300 × 300 read length at the MIT BioMicroCenter sequencing core.

### Statistical analysis of DMS reactivities and structural features

DMS-induced mutation rates (DMS reactivities) were determined using the Detection of RNA folding Ensembles with Expectation Maximization clustering (DREEM) pipeline version 1.0[66], using the default

parameters except for a 90% coverage threshold for clustering. In order to control for variations in DMS treatment among different samples, the DMS reactivities were normalized using a custom script as follows. The median DMS reactivity among the top 50% ($n = 46$) of the 91 adenine (A) and cytosine (C) bases in the spiked-in RRE control was computed for each sample. For each other sample, the ratio of the median DMS reactivity of the RRE to the median DMS reactivity of the RRE in the sample of 23S scaffold without staples (the reference sample) was computed, and the DMS reactivity of the sample was divided by this ratio to normalize it.

We developed the software ARIADNE to determine the locations of these structural features using the outputs from pyDAEDALUS. ARIADNE was written in Python 3.8 and uses NumPy 1.20[98], Pandas 1.2[99], Matplotlib 3.3[100], Seaborn 0.10[101], and BioPython 1.7[102]. For each origami, ARIADNE was used to generate a table of all nucleotides in the scaffold and staple strands, indicating for each nucleotide the identity of its base (i.e. A, C, G, T, or U) and the identity of the adjacent structural feature (if any), which could be one of seven types: double crossover of the scaffold strand (scaffold DX2), 5′ or 3′ terminus of the scaffold stand (scaffold nick), double crossover of the staple strand (staple DX2), single crossover (a.k.a. mesojunction) of the staple strand (staple DX1), 5′ or 3′ terminus of a staple stand that lies on the same double helix as and adjacent to that strand's 3′ or 5′ terminus (staple nick), 5′ or 3′ terminus of a staple strand that abuts a single crossover of another staple strand (staple single term), or a vertex of the polyhedral origami. Scaffold nicks were excluded from further analysis due to insufficient coverage.

ARIADNE was used to identify all double helical segments, defined as a set of contiguous scaffold-staple base pairs bordered by a structural feature on each end with no structural features in the middle, as in a previous work[103]. The DMS reactivities at the terminal scaffold bases in each segment, the mean DMS reactivity over the interior bases in each segment, their ratios, and the DMS reactivities of bases up to 6 nt upstream and downstream of each structural feature were computed and plotted with a custom Jupyter notebook written in Python 3.8, using Pandas 1.2, Matplotlib 3.3, and Seaborn 0.10. The melting temperature of each segment was predicted using a nearest neighbor model[104] and salt correction[105] for RNA/DNA duplexes, with 300 mM Na⁺, implemented in BioPython 1.7. All statistical significance tests and estimates of parameters were performed using SciPy 1.6[106]. Only non-parametric tests were used in order to avoid making assumptions about the underlying distributions. Each analysis was also repeated on the DMS reactivities of only adenines and of only cytosines separately in order to control for the covariate of base identity. A table of all DMS reactivities and structural features of every origami analyzed in this study is given in Supplementary Data 4.

## Cryo-electron microscopy

Three microliters of the folded and purified RNA nanostructure solution (approximately 600 nM) were applied onto the glow-discharged 200-mesh Quantifoil 2/1 grid, blotted for four seconds and rapidly frozen in liquid ethane using a Vitrobot Mark IV (Thermo Fisher Scientific). All grids were screened and imaged on a Talos Arctica cryo-electron microscope (Thermo Fisher Scientific) operated at 200 kV at a magnification of 79,000× (corresponding to a calibrated sampling of 1.76 Å per pixel). Micrographs were recorded by EPU software (Thermo Fisher Scientific) with a Gatan K2 Summit direct electron detector in counting mode, where each image is composed of 24 individual frames with an exposure time of 6 s and a total dose ~63 electrons per Å². We used a defocus range of −1.5−−3 μm to collect images, which were subsequently motion-corrected using MotionCor2[107]; CTF was estimated using CTFFIND4[108]. The initial particles were picked manually by e2boxer.py in EMAN2[109]. The particles were imported into cryoSPARC[110] software for further 2D analysis and 3D refinement. For the A-form origami, the percentages of particles kept from initial

picking and 2D analysis for final reconstructions were approximately 32.8% for rT66, 17.5% for rO44, 46.9% for rO66, and 40.2% for rPB66. The total number of images collected and total particles used for final refinement are listed in Supplementary Table 1. Resolution for the final maps were estimated using the 0.143 criterion of the Fourier shell correlation (FSC) curve without any mask. A Gaussian low-pass filter was applied to the final 3D maps displayed in the UCSF Chimera software package[111]. Correlation of each map with its corresponding atomic model is calculated by the UCSF Chimera fitmap function, with density simulated from the model at the same resolution as the corresponding reconstruction. Density map edge length measurements were performed with the tape measure tool in UCSF Chimera X[112], measuring each edge and averaging the edge length measurements per object. For comparison, each A-form origami object was also reconstructed without symmetry imposed, and these maps are displayed in the supplement (Supplementary Figs. 5e, 6e, 9e, and 10e).

## Generation of De Bruijn sequences for RNA scaffolds

Sequences of De Bruijn scaffolds were designed using a De Bruijn order of 8, such that no sub-sequence of 8 nucleotides was identical to any other sub-sequence of 8 nucleotides or its reverse complement. A single sequence of 1218 nt was generated; the first 660 and 924 nt were used as the scaffolds of rT55 and rT77, respectively.

## Reporting summary

Further information on research design is available in the Nature Portfolio Reporting Summary linked to this article.

## Data availability

The electron density maps from cryo-EM generated in this study have been deposited in the EMDB repository under accession codes EMD-34058, EMD-34059, EMD-34060, and EMD-34061. The sequence alignment maps from DMS-MaPseq generated in this study have been deposited in the NCBI Sequence Read Archive under accession code PRJNA868816. A table of all DMS reactivities and structural features of every origami is provided in Supplementary Data 4. All other data generated or analyzed during this study are included in this article, its Supplementary Information and Supplementary Data files. Source data are provided with this paper.

## Code availability

Source code for pyDAEDALUS is available for download from GitHub at https://github.com/lcbb/pyDAEDALUS. The source code for ARIADNE, used to annotate structural features in the origami, as well as the algorithm for generating De Bruijn scaffold sequences, is available for download from GitHub at https://github.com/lcbb/ariadne. The custom Jupyter notebook for analyzing the DMS-MaPseq data from DREEM is available for download from GitHub at https://github.com/lcbb/3dRnaScaffoldedWireframeOrigami. The source code for the Rouls package (a dependency of the Jupyter notebook) is available for download from GitHub at https://github.com/lcbb/rouls.

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

## Acknowledgements

This work was supported by the National Science Foundation (NSF-EAGER) CCF-1547999 to M.B., the Office of Naval Research (N000141612953, N000141210621, N000141612506) to M.B., a NDSEG fellowship to M.F.P., an NSF GRFP to M.F.A., the National Institute of Environmental Health Sciences (P30-ES002109) and the National Institutes of Health (1-R21-EB026008-01). We thank Tammy C. T. Lan for assistance with DMS-MaPseq, William Bricker for assistance with the PDB generation portion of pyDAEDALUS, Kaitlin Tucci for assistance with initial manual designs, and Hellen Huang for running the enzyme stability assay. We also thank George Church for the suggestion to use ribosomal RNA as an abundant origami scaffold source. Molecular graphics and analyses were performed with UCSF Chimera and UCSF ChimeraX, developed by the Resource for Biocomputing, Visualization, and Informatics at the University of California, San Francisco, with support from NIH P41-GM103311, NIH R01-GM129325 and the Office of Cyber Infrastructure and Computational Biology, National Institute of Allergy and Infectious Diseases. Some cryo-EM specimens were prepared and imaged at the Automated Cryogenic Electron Microscopy Facility in MIT.nano on a Talos Arctica microscope, which was a gift from the Arnold and Mabel Beckman Foundation.

## Author contributions

T.R.S. and M.F.P. conceived of the sequence design approach for A-form DX wireframe 3D origami. T.R.S. developed initial manual designs. T.R.S., S. Ratanalert, and M.F.P. implemented the design algorithm in the Python-adapted version of DAEDALUS and T.R.S. built the graphical user interface. M.F.P. and T.R.S. designed and implemented the experimental folding assays. M.F.A., M.F.P. and S. Rouskin designed the DMS-MaPseq experiments. M.F.A. collected and analysed the DMS-MaPseq data. K.M.P. and M.F.P. collected the dynamic light scattering data. W.C., S.L., and K.Z. contributed to the cryo-EM experimental design. M.F.P. and S.L. collected the cryo-EM images, for subsequent reconstruction by S.L. M.B. supervised the project. M.F.P., M.F.A., and T.R.S. wrote the manuscript. All authors discussed the results and edited the manuscript.

## Competing interests

M.B., T.R.S., and M.F.P. are co-inventors on a patent pending (U.S. Patent Application No. 63/324,538) for the A-form routing scheme employed for 3D RNA-scaffolded DX wireframe origami. The remaining authors declare no other competing interests.
