## [Peer Review File · Nature Communications]

3D RNA-scaffolded wireframe origamiREVIEWER COMMENTS

Reviewer #1 (Remarks to the Author):

Parsons et al describe wireframe origami objects folded from RNA scaffolds and DNA staples. Different sources of RNA scaffolds were used (natural origin, synthetic, and mRNA coding for EGFP) and investigated for their ability to fold into target objects when folded with DNA staples. Design rules were investigated to understand to which extent A- or B-Form helical geometry aspects need to be used for successful assembly. The daedalus software was adapted to allow users to make such objects.

Assembly- and structure validation was carried out meticulously using gel electrophoresis, chemical foot printing. and in great depth using cryo EM. The data presented is convincing. A lot work went into this manuscript. In my view all conclusions are fully supported by data, this paper is a valuable and important contribution to the field that opens new vistas for applications, such as cytosolic delivery of coding RNA scaffolds for gene expression among others. I recommend publication.

Two things that I noticed: ref 41 with Alexander Rich's ref has "PUBLISHED" in the title, which is probably a typo? Then, some of the early gel images shown in the supplement could potentially be improved and made more informative, as the bands of interest are often quite faint. But this is a really minor issue actually.

Reviewer #2 (Remarks to the Author):

The manuscript entitled "3D RNA-scaffolded wireframe origami" by Parsons et al. describes a study of the design and characterization of hybrid RNA:DNA wireframe origami, where RNA of biological origin such as mRNAs and rRNAs are used as scaffolds. The study uses the DAEDALUS software to design RNA:DNA wireframe origamis and characterizes the structures by gel electrophoresis, DMS probing and cryo-EM. The study also for the first time probes the secondary structure of origami structures using DMS-MaPseq, which is made possible by the RNA:DNA hybrid structures. The study is in general well-documented and carefully executed. The results are noteworthy because of the generality of the approach and because of the promising applications in nanomedicine. The cryo-EM characterization, however, leaves me with a bit doubt about the yield and the precise geometrical shape of the 3-dimensional structures that I think should be addressed before publication.

Major comments:

(1) The cryo-EM 3D reconstructions are made with symmetry imposed, which I think gives a slightly wrong impression of the variability of the actual particles. Could the authors show the 3D reconstructions without symmetry in the supplementary material, so that the reader can also get a more realistic view of the actual shapes?

(2) From looking at the original cryo-EM images it seems that there is more diversity of shapes than revealed by the shown 2D class averages and the 3D reconstruction with symmetry imposed. It would be valuable to estimate what fraction of the particles adopt the ideal shape versus the rest of the particle population, which would also be important for future use in medicine. In the materials and methods, it is described that particles were picked manually. Could the authors do a more unbiased automated particle picking and evaluate (and show) 2D class averages to provide more insight into the diversity of shapes? Recent cryo-EM software packages allow 3D variability analysis (e.g. cryoSPARC), which could be a suggestion for evaluating the diversity observed. The flexibility at vertices and crossovers could possibly help explain the increased DMS reactivities observed at these sites.

Minor comment:

(1) In some of the references I found a formatting error, where several papers are referred to as "Science (1979)".

RESPONSE TO REFEREES

Reviewer #1 (Remarks to the Author):

Parsons et al describe wireframe origami objects folded from RNA scaffolds and DNA staples. Different sources of RNA scaffolds were used (natural origin, synthetic, and mRNA coding for EGFP) and investigated for their ability to fold into target objects when folded with DNA staples. Design rules were investigated to understand to which extent A- or B-Form helical geometry aspects need to be used for successful assembly. The daedalus software was adapted to allow users to make such objects.

Assembly- and structure validation was carried out meticulously using gel electrophoresis, chemical footprinting, and in great depth using cryo EM. The data presented is convincing. A lot of work went into this manuscript. In my view all conclusions are fully supported by data, this paper is a valuable and important contribution to the field that opens new vistas for applications, such as cytosolic delivery of coding RNA scaffolds for gene expression among others. I recommend publication.

Response: We thank the reviewer for their positive evaluation of the paper.

Two things that I noticed: ref 41 with Alexander Rich's ref has "PUBLISHED" in the title, which is probably a typo?

Response: We thank the reviewer for pointing out this typo in the references section. We have removed the word "PUBLISHED" from the title of ref 41.

Then, some of the early gel images shown in the supplement could potentially be improved and made more informative, as the bands of interest are often quite faint. But this is a really minor issue actually.

We appreciate that some gel electrophoresis bands in the supplement were faint and would benefit from improvement. We repeated the gel for Supplementary Fig. 2 with greater loading and higher-gain imaging to improve the visibility of the bands, and replaced the gel image in Supplementary Fig. 2 accordingly:

Reviewer #2 (Remarks to the Author):

The manuscript entitled “3D RNA-scaffolded wireframe origami” by Parsons et al. describes a study of the design and characterization of hybrid RNA:DNA wireframe origami, where RNA of biological origin such as mRNAs and rRNAs are used as scaffolds. The study uses the DAEDALUS software to design RNA:DNA wireframe origamis and characterizes the structures by gel electrophoresis, DMS probing and cryo-EM. The study also for the first time probes the secondary structure of origami structures using DMS-MaPseq, which is made possible by the RNA:DNA hybrid structures. The study is in general well-documented and carefully executed. The results are noteworthy because of the generality of the approach and because of the promising applications in nanomedicine. The cryo-EM characterization, however, leaves me with a bit doubt about the yield and the precise geometrical shape of the 3-dimensional structures that I think should be addressed before publication.

Major comments:

(1) The cryo-EM 3D reconstructions are made with symmetry imposed, which I think gives a slightly wrong impression of the variability of the actual particles. Could the authors show the 3D reconstructions without symmetry in the supplementary material, so that the reader can also get a more realistic view of the actual shapes?

Response: We have added SI figure panels to show the 3D reconstructions without symmetry, as suggested (Supplementary Fig. 5e, 6e, 9e, and 10e):

(2) From looking at the original cryo-EM images it seems that there is more diversity of shapes than revealed by the shown 2D class averages and the 3D reconstruction with symmetry imposed. It would be valuable to estimate what fraction of the particles adopt the ideal shape versus the rest of the particle population, which would also be important for future use in medicine. In the materials and methods, it is described that particles were picked manually. Could the authors do a more unbiased automated particle picking and evaluate (and show) 2D class averages to provide more insight into the diversity of shapes? Recent cryo-EM software packages allow 3D variability analysis (e.g. cryoSPARC), which could be a suggestion for evaluating the diversity observed. The flexibility at vertices and crossovers could possibly help explain the increased DMS reactivities observed at these sites.

Response: As noted by the reviewer, numerous bad particles were removed during particle picking and 2D analysis. The fractions of good particles suitable for reconstruction were approximately 17.5% for rO44, 46.9% for rO66, 32.8% for rT66, and 40.2% for rPB66, respectively, which we have added to the revised manuscript.

We thank the reviewer for their suggestion on particle picking. We actually did try using several different software packages during data processing, however, it was difficult to apply automated particle picking to achieve well-boxed particles, possibly because these RNA objects are not sufficiently compact.

We also thank the reviewer for their suggestion regarding 3D variability analysis. We performed this analysis using cryoSPARC on rO44 as an example to visualize flexibility. We used “cluster” as the output mode, which resulted in four 3D classes. There were not significant differences visible amongst these 3D classes (see the images below) except for minor variance at vertices and crossovers, which might suggest that our RNA scaffolded objects were relatively stable.

Minor comment:

(1) In some of the references I found a formatting error, where several papers are referred to as “Science (1979)”.

Response: We thank the reviewer for identifying this formatting error in the references; we have corrected the Journal name in these references to “*Science*” accordingly.

REVIEWERS' COMMENTS

Reviewer #2 (Remarks to the Author):

The authors have responded well to my concerns about the effect of imposing symmetry in the 3D reconstructions, the yield of well-formed particles, and the possible flexibility of structures as studied by 3D variability analysis. I can therefore fully recommend publication of the paper.